# The relationship between physicians' self-kindness and professional fulfillment and the mediating role of personal resilience and work-home interference: A cross-sectional study

Rosa Bogerd[1,2]*, Maarten P. M. Debets[1,2], Debby G. Keuken[3‡], Rutger J. Hassink[4‡], José P. S. Henriques[3,5‡], Kiki M. J. M. H. Lombarts[1,2]

1 Department of Medical Psychology, Professional Performance & Compassionate Care Research Group, Amsterdam UMC Location University of Amsterdam, Amsterdam, The Netherlands, 2 Amsterdam Public Health Research Institute, Amsterdam, The Netherlands, 3 The Netherlands Society of Cardiology, Utrecht, The Netherlands, 4 Department of Cardiology, University Medical Center Utrecht, Utrecht, The Netherlands, 5 Department of Cardiology, Amsterdam UMC Location University of Amsterdam, Amsterdam, The Netherlands

☯ These authors contributed equally to this work.
‡ DGK, RJH and JPSH also contributed equally to this work.
* r.bogerd@amsterdamumc.nl

**Data Availability Statement:** The data underlying this article are partly owned by the NVVC.

## Abstract

### Background

Professional fulfillment is crucial for physicians' well-being and optimal patient care. Highly demanding work environments, perfectionism and self-critical attitudes jeopardize physicians' professional fulfillment.

### Objective

To explore to what extent a kinder attitude towards the self, i.e. self-kindness, was associated with physicians' professional fulfillment and whether this relationship was mediated by personal resilience and work-home interference.

### Methods

In 2020, cardiologists (*n* = 374) in the Netherlands participated in a web-based survey. Self-kindness was measured by the self-kindness subscale of the Self-Compassion Scale, personal resilience by the Brief Resilience Scale, work-home interference by the negative Work-Home Interference subscale of the Survey Work-Home Interaction–Nijmegen, and professional fulfillment by the corresponding subscale of the Professional Fulfillment Index. Using Hayes' SPSS macro PROCESS v3.5, the authors tested the parallel mediation model.

Predetermined agreements in the contract between both parties create legal restrictions on sharing the data. Data will be shared on reasonable request to the NVVC (bureau@nvvc.nl) or to the corresponding author (r.bogerd@amsterdamumc.nl) with permission of the Netherlands Society of Cardiology.

**Funding:** This research was initiated and partly funded by the Netherlands Society of Cardiology (NVVC - https://www.nvvc.nl/), award number N/A. Three of the authors (DK, RJ and JH) are related to the NVVC. They participated in designing the study, collecting and analysing the data, preparing the manuscript and agreed with the decision to publish the manuscript.

**Competing interests:** I have read the journal's policy and the authors of this manuscript have the following competing interests: co-author Debby G. Keuken is a senior policy officer at the NVVC. Co-author Rutger J. Hassink was chair of the Quality Committee of the NVVC during the period of data-collection. Co-author José P. S. Henriques is a board member of the NVVC. The NVVC commissioned this project and was a party in the data agreements. In agreement with the NVVC, the authors have decided to make the data available upon request. The other authors have no conflicts of interest.

## Results

Self-kindness was not directly associated with professional fulfillment (direct effect = .042, $p$ = .36, 95% CI: -0.048, 0.132). Self-kindness was indirectly related to professional fulfillment through individual resilience (indirect effect = .049, 95% CI: .020, 0.086) and work-home interference (indirect effect = .057, 95% CI: .023, 096).

## Conclusions

This study suggests that improving physicians' self-kindness may enhance professional fulfillment through personal resilience and work-home interference. Our findings may stimulate and remind physicians to be kind towards themselves as it may benefit them and their patients.

## Introduction

Professionally fulfilled physicians derive intrinsic positive rewards from their work, such as professional satisfaction, self-efficacy and happiness. As they work with joy and experience meaningfulness, they are less prone to burnout and leaving the profession than unfulfilled physicians. Patients of professionally fulfilled physicians report higher levels of satisfaction, whereas low professional fulfillment is associated with an increased chance on medical errors [1–5]. Unfortunately, physicians' professional fulfillment is under pressure due to demanding work environments and unhealthy norms in the medical professional culture [1, 6, 7]. While improving physicians' work environments has been high on the agenda of healthcare organizations and professional bodies, work pressures remain high [8]. Unhealthy norms like working late, deprioritizing self-care and shaming approaches after (near-)mistakes neither are supportive for physicians' professional fulfillment [6, 7, 9]. On top of that, in striving to deliver high-quality patient care physicians often develop perfectionistic mindsets, a self-critical attitude and low self-tolerance [6, 7, 9, 10]. Both perfectionism and self-criticism have been associated with burnout in physicians [11–13]. A kinder attitude towards the self, i.e. self-kindness, has been found to buffer against stress and burnout [14–18].

Self-kindness refers to being gentle, supportive and understanding towards the self, instead of being harsh and self-critical [19]. Research has shown that it can be trained in healthcare professionals [9, 18]. While ample evidence shows that self-kindness is associated with reduced burnout in physicians [14–17], coming from a positive psychology perspective [20], we argue that physician well-being encompasses more than the absence of burnout. It is also worthwhile to explore whether kinder self-attitudes are to increase positive well-being indicators such as professional fulfillment, given its presumed benefits for physicians and their patients. However, the relationship between self-kindness and professional fulfillment thus far remains largely underexplored. Previous research has shown how physicians' self-critical and perfectionistic attitudes may harm their professional fulfillment [6, 7], but research on how opposite self-attitudes, such as self-kindness, may actually benefit it, is scarce. Empirical research on self-kindness as a potential predictor for professional fulfillment will add to the body of literature on physician well-being from a more positive angle.

Indications for a direct relationship between self-kindness and professional fulfillment are studies showing that self-kindness stimulates intrinsic motivation, goal-setting behaviors and feelings of happiness and connectedness [21–23], all core characteristics of professional

fulfillment [24]. For indirect relationships, the current literature suggests two worthwhile constructs to explore. First, researchers have found positive relationships between self-kindness and personal resilience–the ability to bounce back [25, 26]. Second, self-kindness positively affects self-care and behavior that stimulates well-being [27–30]. Self-kindness seems beneficial for physicians as they often struggle with managing their work-home balance [31]. While both being crucial for maintaining and enhancing the quality of patient care [5, 32, 33], higher levels of personal resilience and fewer work-home interferences have also been found to positively associate with work engagement and job satisfaction amongst physicians [34–38]. Professional fulfillment is an indicator of occupational well-being, like work engagement and job satisfaction, but differs from these constructs by also including a sense of meaningfulness and personal growth [24]. Based on the evidence aforementioned, we hypothesize that i) self-kindness is directly related to professional fulfillment, as well as ii) indirectly, through personal resilience and work-home interference.

This study focuses on cardiologists, as the profession of cardiology is known for its competitive culture stimulating its members to appear impervious to pressure- raising worries about their occupational well-being [39, 40]. This study aims to contribute to the literature and practice by addressing the following research questions: To what extent 1) is cardiologists' self-kindness related to their professional fulfillment? and 2) is this relationship mediated by personal resilience and work-home interference? The outcomes of this study provide more insight into cardiologists' attitudes towards the self and how they relate to their occupational well-being. Taking into account the aforementioned detrimental aspects of medicine's professional culture, a better understanding of the role and impact of self-kind attitudes may offer a first step in finding ways to redefine and improve practice [6, 7]. The results of this study may inform new approaches to enhance physicians' professional fulfillment and, subsequently, patient care. In the next section we discuss our research methods, followed by the presentation of our results, a discussion of our findings and a conclusion.

## Methods

### Study design, population and setting

This cross-sectional survey study was conducted in 2020 amongst all registered cardiologists who were a member of the Netherlands Society of Cardiology (NVVC) and were working in the Netherlands. In the Netherlands in 2021 approximately 48% of the cardiologists was self-employed and 52% was employed by hospitals- they deliver their medical services within general hospitals, top clinical hospitals, university hospitals and independent treatment centers (unpublished information of the NVVC). In 2021, the average age of cardiologists in the Netherlands was 48 years and approximately 72% of all cardiologists were male. As a profession, cardiology is known to be demanding and strenuous [39, 41].

### Measures

We used validated measures for all included variables and, when needed, four researchers independently translated English measurements to Dutch using the forward-backward translation method [42]. We measured self-kindness using the 5-item self-kindness domain of the self-compassion scale (SCS) developed by Kristin Neff [19]. This domain measures "how often people respond to feelings of inadequacy or suffering" with self-kindness versus with self-judgment [19]. Examples of self-kindness items are "I try to be loving toward myself when I'm feeling emotional pain" and "I'm tolerant of my own flaws and inadequacies." Respondents provided their answers on a 5-point Likert scale from 'almost never' to 'almost always' [19]. The reliability coefficient using Cronbach's alpha for this scale was 0.82 in our sample.

We measured personal resilience using the 6-item Brief Resilience Scale (BRS) developed by Smith and colleagues [43]. Respondents provided their answers on a 5-point Likert scale ranging from 'strongly disagree' to 'strongly agree.' Three items are positively worded, i.e., "I tend to bounce back quickly after hard times", while the other 3 items are negatively worded, as in "It is hard for me to snap back when something bad happens" [43]. The reliability coefficient using Cronbach's alpha for this scale was 0.86 in our sample.

We assessed work-home interference using the 9-item sub-scale negative Work-Home Interference (WHI) from the Survey Work-Home Interaction–Nijmegen (SWING), which measures the level at which someone's work negatively influences someone's functioning at home [44]. Items are scored on a 4-point Likert scale 'how often' scale ranging from '(practically) never' to '(practically) always.' Examples of negative WHI items are "How often do you not fully enjoy the company of your spouse/ family/ friends because you worry about your work?" and "How often do your work obligations make it difficult for you to feel relaxed at home?" [44] The reliability coefficient using Cronbach's alpha for this scale was 0.88 in our sample.

We measured professional fulfillment using the 6-item professional fulfillment scale from the Professional Fulfillment Index (PFI) developed by Trockel et al. [24] The scale measures the intrinsic positive rewards an individual receives from doing her or his work, including items about meaningfulness, contribution, happiness, satisfaction, self-worth and feeling in control when dealing with problems. Items are scored on a 5-point Likert scale with options ranging from 'totally disagree' to 'totally agree' [24]. Usually, the PFI measures professional fulfilment on a scale from 0 to 4. In order to guarantee consistency between the scales of the four main variables we measured professional fulfillment on a scale from 1 to 5, as used for the other scales. Example items of the professional fulfillment scale are: "My work is satisfying to me" and "I'm contributing professionally (e.g. patient care, teaching, research and leadership) in ways that I value most." [24] The reliability coefficient using Cronbach's alpha for this scale was 0.88 in our sample.

## Data collection

This study was part of a larger data collection about the occupational well-being of Dutch cardiologists, commissioned by the Netherlands Society of Cardiology (NVVC). On September 28, 2020, on behalf of the NVVC, the researcher (RB) invited all cardiologists practicing in the Netherlands and registered as an NVVC member, individually per email to participate in an online questionnaire, using Castor EDC. Participation was voluntary. Digital informed consent was obtained and participants' anonymity and confidentiality were safeguarded. Apart from occupational well-being related items, the questionnaire also included questions on respondents' demographics, i.e. sex (male/ female), age (years), years of experience as a cardiologist (year of first registration) and type of institution where the medical specialist was working (general hospital, top clinical hospital, university hospital, independent sector treatment center). These four demographics are able to provide a basic description of our sample and were considered most relevant in explaining physicians' occupational wellbeing by the researchers. Additionally, the NVVC deemed them potentially relevant for defining new, tailored wellbeing policies or interventions. The survey took place from September 28 till December 6, 2020; up till four reminders were sent during the survey period. We have included an appendix with the survey items relevant for the current study as S1 Appendix.

## Statistical analyses

We included cardiologists who reported scores on (the separate items of) all main variables: (1) self-kindness, (2) personal resilience, (3) work-home interference and (4) professional fulfillment. Data were screened for extreme or unrealistic scores using the SPSS functions 'Sort ascending' and 'Sort descending'. No extreme or unrealistic scores were found in the data, leaving 374 cases for analysis.We rightly expected that our data collection would result in a sample that exceeds the general rule of thumb for minimum sample sizes in regression analyses (of which our parallel mediation model consists) [45, 46].) Therefore, we would have sufficient power to detect significant associations between the variables under study and we did not perform an a priori power analysis. With the cleaned data, scale scores for the variables were computed by averaging the individual items. Next, we used descriptive statistics to describe our sample characteristics and main variables.

To answer our research question "to what extent is physicians' self-kindness related to their professional fulfillment, and is this relationship mediated by personal resilience and work-home interference?", we performed a parallel mediation analysis with Hayes' SPSS macro PROCESS v3.5 [47]. Hayes' macro for SPSS is a widely used tool to test parallel mediation models [48]. The procedure makes use of bootstrapping technique to determine the indirect effects. Compared to the Sobel test, which is an alternative way for determining the effects of mediating variables, bootstrapping requires fewer assumptions such as no need for normality in the distributions of the variables, provides a higher study power and lowers the risk of falsely rejecting the null hypothesis [48].

We checked the statistical assumptions for parallel mediation of linearity, homoscedasticity and normality of estimation error by plotting the standardized residuals of all separate regressions building the parallel mediation model [47]. We found no worrying violations of the statistical assumptions. Additionally, we used Pearson's Correlations for calculating the associations among the main variables to check for multicollinearity. We then tested our parallel mediation model with self-kindness as the independent variable, personal resilience and work-home interference as mediators and professional fulfillment as the dependent variable. Preliminary univariate analyses (T-tests and ANOVA's) found no statistically significant relationship of any of the demographic variables with professional fulfillment. In line with other well-being research [49–51], however, sex and years of registration were included as covariates. Because of the multicollinearity of age and years of registration (r = -.824, p < 0.001) [52], we opted to use the latter in the analysis as it was considered more clinically relevant; the year in which an individual first registered as a medical specialist better reflects their experience as a practicing physician than their age does. We generated a confidence interval (95% CI) from 10000 resamples to examine the significance of our indirect effects. The number of 10000 resamples is recommended by Hayes [47]. All analyses were performed using the SPSS Statistics (version 26; IBM, Armonk, New York).

## Ethical approval

The institutional ethical review board of the Amsterdam UMC (METC) of the University of Amsterdam provided a waiver declaring the Medical Research Involving Human Subjects Act (WMO) did not apply to the current study (reference number W20_324 # 20.323). Written informed consent was obtained in the online survey. Participants were only allowed to continue the survey when they entered 'Yes' for "I agree to participate in this study". Information about the research was sent to the participants via email previously and again provided at the beginning of the survey.

## Results

Table 1 reports the demographic characteristics of the respondents in this study. In total, 374 (response rate of 34.6%) Dutch cardiologists completed the questionnaire. Most respondents were male ($n$ = 273, 73.0%), and the most frequently rated age category and hospital type were 36–45 years old ($n$ = 145, 38.8%) and general hospital ($n$ = 154, 42.2%). The results are not too deviant from the average age of cardiologists in the Netherlands (almost 70% of our sample was between 36 and 55 years old) and the percentage of male cardiologists (72.0%); for type of hospital and years of experience no population numbers were available. Eight cases were excluded due to incomplete scores on the main variables.

Table 2 reports the mean scores and standard deviations of the four main variables and the cross-variables' associations. On a 5-point Likert scale respondents' mean scores were 2.76 (SD = 0.7) for self-kindness, 3.85 (SD = 0.6) for professional fulfillment and 3.46 (SD = 0.5) for personal resilience. For work-home interference, respondents reported an average mean score of 2.23 (SD = 0.5) on a 4-point Likert scale.

Fig 1 shows the unstandardized regression coefficients of the parallel mediation model. The results show a significant indirect effect of self-kindness on professional fulfillment through both personal resilience (indirect effect (a1b1) = .049, 95% CI: .020, .086) and work-home interference (indirect effect (a2b2) = .057, 95% CI: .023, .096). The combined indirect effect of personal resilience and work-home interference was also significant (indirect effect (total) = .106, 95% CI .061, .156). No significant direct effect of self-kindness on professional fulfillment was found (direct effect (c') = .042, $p$ = .36, 95% CI: -0.048, 0.132). A negligible but significant positive effect was found for first registration year on work-home interference ($b$ = .008, $p$ = .007, 95% CI: 0.002–0.014).

**Table 1. Sample characteristics of 374 registered cardiologists in the Netherlands in 2021.**

| Demographic characteristics | N (374) | Frequency (%) |
|---|---|---|
| Sex | | |
| Male | 273 | 73.0 |
| Female | 101 | 27.0 |
| Age (years) | | |
| < 36 | 15 | 4.0 |
| 36–45 | 145 | 38.8 |
| 46–55 | 112 | 29.9 |
| 56–65 | 95 | 25.4 |
| > 65 | 7 | 1.9 |
| First registration (year) | | |
| 1981–1990 | 19 | 5.1 |
| 1991–2000 | 64 | 17.1 |
| 2001–2010 | 139 | 37.2 |
| 2011–2021 | 152 | 62.8 |
| Type of institute | | |
| General hospital | 154 | 41.2 |
| Tertiary referral hospital | 129 | 34.5 |
| University hospital | 62 | 16.6 |
| Independent sector treatment center | 20 | 5.5 |
| Missings | 9 | 2.4 |

**Table 2. Means, standard deviations and pearson correlations for self-kindness, professional fulfillment, personal resilience and work-home interference.**

| Variables (scale) | Mean | SD | 1 | 2 | 3 |
|---|---|---|---|---|---|
| 1. Self-kindness (1–5) | 2.76 | 0.68 | | | |
| 2. Professional fulfilment (1–5) | 3.85 | 0.62 | 0.17** | | |
| 3. Personal resilience (1–5) | 3.46 | 0.67 | 0.24** | 0.32** | |
| 4. Work-home interference (1–4) | 2.23 | 0.51 | -0.26** | -0.35** | -0.37** |

** (p < 0.01), n = 374.

## Discussion

### Main findings

This study explored the relationship between self-kindness and professional fulfillment amongst cardiologists and whether this relationship was mediated by personal resilience and work-home interference. In contrast to our first hypothesis, the results did not show a direct relationship between self-kindness and professional fulfillment. However, as hypothesized, this study showed that cardiologists who were more self-kind reported higher levels of personal resilience and less work-home interference, which in turn benefited their professional fulfillment.

### Explanation of findings

The average professional fulfillment score of Dutch cardiologists (M = 3.85) is relatively high, compared to those of physicians in general which (converted to our measurement scale) have been reported between 3.25 and 3.50 [24, 53]. One explanation for this finding may lie in the highly rewarding and meaningful nature of the specialty–cardiologists are frequently present at important moments in patients' lives which may be a stimulator for their feelings of intrinsic rewards and motivation [1, 39, 54]. Factors that have been found to positively affect the professional fulfillment of Dutch cardiologists are indeed satisfying physician-patient interactions, but also autonomy, personal resilience, satisfaction with compensation levels and the degree to which the job matches an individual's energy resources [54]. This study adds to that by showing how self-kindness indirectly affects cardiologists' professional fulfillment. Previous research has shown that patients of professionally fulfilled physicians are more satisfied [1]

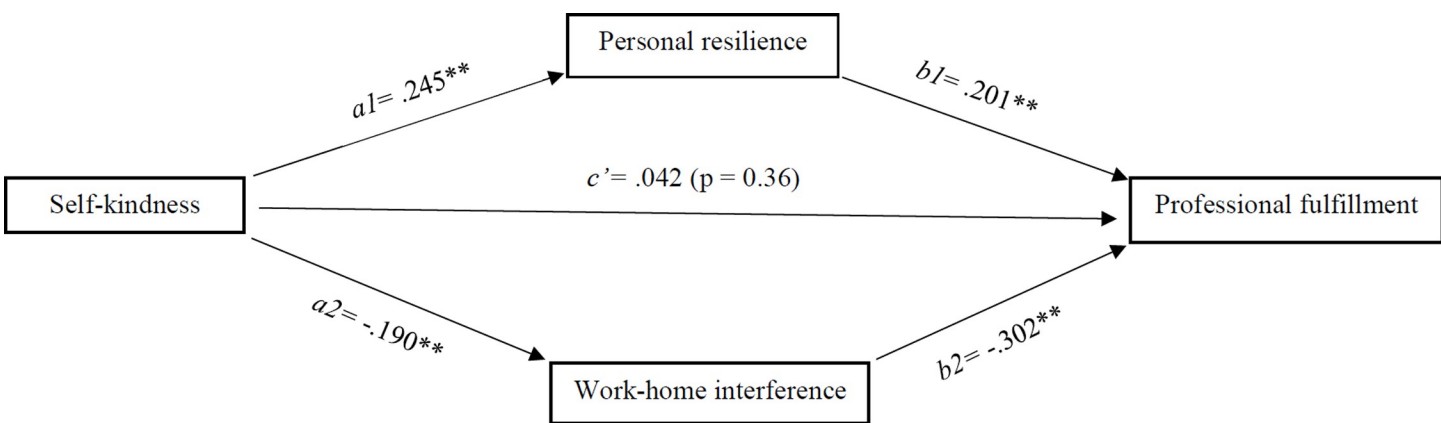

**Fig 1. Conceptual model.** The model depicts the relationship between self-kindness and professional fulfillment, mediated by personal resilience and work-home interference, found amongst cardiologists in the Netherlands (n = 374) in 2020. The model was adjusted for sex and years of registration. **(p < 0.01).

and that self-kindness has been found to be a prerequisite for compassionate care [18, 55]. Therefore, enhancing cardiologists' self-kindness may benefit patients in multiple ways.

Dutch cardiologists reported an average self-kindness score of 2.76 on a 5-point Likert scale. While there is little comparable evidence on physicians' self-kindness specifically, some studies do report physicians' self-compassion scores, which include the subscales of self-kindness, mindfulness and common humanity [14, 56]. Physicians in Canada and New Zealand report mean self-compassion scores of 3.4 and 3.2. According to Neff, leading expert on self-compassion, people scoring between 2.5 and 3.5 on the self-compassion scale can be considered moderately self-compassionate [57]. Generally, individuals score lowest on the self-kindness subscale of the self-compassion scale [58]. Dutch cardiologists' can be considered moderately self-compassionate, assuming that the scores on the other subscales are slightly higher than 2.74. Still, there is substantial room for improvement.

According to Shanafelt et al. [7], medicine's professional culture lacks self-compassion and creates perfectionistic and self-critical individuals who normalize deferring self-care and personal relationships. Researchers argue that physicians report lower levels of self-compassion than other healthcare professionals, e.g. nurses, due to the way self-compassion is seen and valued within their profession [14]. As the profession of cardiology is known to be competitive, its members often develop perfectionist mindsets [39, 40] and frequently face critical or confronting situations [39], the found moderate score on self-kindness amongst Dutch cardiologists indicates that they, and their patients, could indeed benefit from being more tolerant towards their own flaws and imperfections.

In contrast to our literature-based expectations, we did not find a direct relationship between cardiologists' self-kindness and professional fulfillment [21–23]. We can think of one theoretical and one methodological potential explanation for the absence of a direct relationship. From a theoretical viewpoint, we regarded self-kindness as the opposite of self-criticism, which is known to potentially harm physicians' professional fulfillment [11–13]. Consequently, we inferred that self-kindness would benefit professional fulfillment. The findings do not confirm this assumption and might imply that self-kindness is not the opposite of self-criticism, or at least does not have the opposite effect on physicians' professional fulfillment. A plausible methodological explanation is that self-kindness relates to specific aspects of the multifactorial professional fulfillment construct only. Being self-kind might not provide meaning in work–a distinctive element of professional fulfillment [24]. Additional analysis of our data seems to support this idea, as self-kindness showed small correlations with all professional fulfillment items except for 'My work is meaningful to me.' Others have found that self-kindness may positively affect certain aspects of professional fulfillment, e.g. feelings of happiness [21–23], and that it may serve as a buffer and burnout [14–16]- the other dimension measured by Trockel et al.'s Professional Fulfillment Index (PFI) [24].

In line with our expectations, this study furthermore shows that self-kindness may indirectly enhance professional fulfillment through personal resilience and work-home interference. Self-kindness may benefit physicians' resilience when they are facing problems or when they are guiding ill patients, implying that it takes less time for them to recover from adverse experiences or confronting situations. Resilient physicians may find more time and space to enjoy positive feelings and opportunities at work, important indicators of professional fulfillment [24]. Research on physician resilience often stresses the influence of genetics, personality traits and an individual's (professional) environment on one's resilience [59, 60]–at times neglecting the possibility that trainable skills or attitudes may also boost individuals' resilience. Our results add to the body of literature [61–63] which demonstrates that trainable skills, such as self-kindness, may also enhance physician resilience. The finding that work-home interference mediates the relationship between self-kindness and professional fulfillment suggests that

self-kind physicians improve their professional fulfillment by better managing their work-home balance. Previous research already showed that fewer work-home interferences positively associate with job satisfaction [36]. Our study supports the body of knowledge showing that self-kindness is associated with self-care skills [27–30], as is seems that better self-care skills could indeed result in more effectively reducing work's negative impact on physicians' personal spheres.

## Strengths and limitations

The strength of this study is that it was set up as a national survey on behalf of the Netherlands Society of Cardiology, thereby inviting all cardiologists registered as a member of the professional body. However, when interpreting the results of this study, two limitations should be considered. First, due to its cross-sectional nature we cannot determine causality. Second, this study may be biased by its relatively low response rate of 34.6%. Nevertheless, response rates of approximately 35% are not uncommon for research into well-being and professional fulfillment amongst medical specialists, especially not during the COVID-pandemic [64, 65]. Further, we do feel reassured by the fact that this study's sample demographic characteristics resemble those of the population of all Dutch cardiologist as reported in unpublished documents of the NVVC.

## Implications for practice and research

Research has shown that self-kindness is a trainable skill, for example, through regular self-kindness exercises. Often self-kindness is embedded as one of the three components of self-compassion, next to common humanity and mindfulness, in self-compassion training programs like Neff et al.'s Mindful Self-Compassion Program [9, 66]. The effectiveness of such programs for physicians' occupational well-being has been demonstrated [15, 67]. This study points to aspects that may be enhanced by the self-kindness elements of these training programs specifically: personal resilience, work-home balance and indirectly physicians' professional fulfillment. Physicians and healthcare organizations responsible for the well-being of the medical workforce may include such programs in their well-being enhancing strategies. More so, the results of this study seem to suggest that physicians operating in demanding work environments with a focus on efficiency, productivity and competitiveness [6, 7] may comfortably be more self-kind instead of being harsh and judgmental towards themselves. Doing so will likely benefit their own well-being and thereby ultimately optimize the quality of patient care.

As this study focused on one component of self-compassion, self-kindness, future research may also include the other two subscales of self-compassion–mindfulness and common humanity–to explore their (combined) effect on professional fulfillment. Also it would be worthwhile to explore whether other work-related aspects or physicians' individual characteristics may mediate the relationship between self-kindness and professional fulfillment. Finally, future research could be aimed at investigating which, if any, self-attitudes may directly enhance physicians' professional fulfillment.

## Conclusions

This was the first study, to our knowledge, to empirically examine the relationship between physicians' self-kindness and professional fulfillment. The results of this study indicate that Dutch cardiologists report relatively high professional fulfillment levels and are moderately self-kind. Cardiologists who perceived themselves as more self-kind did not report higher levels of professional fulfillment. However, they reported higher levels of resilience and a better

work-home balance, which positively contributed to their professional fulfillment. This study may stimulate and remind physicians to be kind towards themselves as it may benefit them and their patients.

## Supporting information

**S1 Appendix. NVVC survey on cardiologists' occupational well-being.** Translated from Dutch to English.
(DOCX)

## Acknowledgments

The authors wish to thank the Netherlands Society of Cardiology for initiating and supporting this research.

## Author Contributions

**Conceptualization:** Rosa Bogerd, Maarten P. M. Debets, Debby G. Keuken, Rutger J. Hassink, José P. S. Henriques, Kiki M. J. M. H. Lombarts.

**Data curation:** Rosa Bogerd, Debby G. Keuken, Rutger J. Hassink.

**Formal analysis:** Rosa Bogerd, Maarten P. M. Debets.

**Funding acquisition:** Debby G. Keuken, Rutger J. Hassink.

**Investigation:** Rosa Bogerd, Maarten P. M. Debets, Kiki M. J. M. H. Lombarts.

**Methodology:** Rosa Bogerd, Maarten P. M. Debets, Debby G. Keuken, Rutger J. Hassink, José P. S. Henriques, Kiki M. J. M. H. Lombarts.

**Project administration:** Rosa Bogerd, Debby G. Keuken, Kiki M. J. M. H. Lombarts.

**Resources:** Kiki M. J. M. H. Lombarts.

**Software:** Rosa Bogerd, Maarten P. M. Debets.

**Supervision:** José P. S. Henriques, Kiki M. J. M. H. Lombarts.

**Visualization:** Rosa Bogerd, Maarten P. M. Debets, Kiki M. J. M. H. Lombarts.

**Writing – original draft:** Rosa Bogerd, Maarten P. M. Debets, Kiki M. J. M. H. Lombarts.

**Writing – review & editing:** Rosa Bogerd, Maarten P. M. Debets, Debby G. Keuken, Rutger J. Hassink, José P. S. Henriques, Kiki M. J. M. H. Lombarts.

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
