## [Decision Letter · Decision Letter 0]

6 Feb 2023

PONE-D-22-32505The relationship between physicians’ self-kindness and professional fulfillment and the mediating role of personal resilience and work-home interference: A cross-sectional studyPLOS ONE

Dear Dr. Bogerd,

Thank you for submitting your manuscript to PLOS ONE. After careful consideration, we feel that it has merit but does not fully meet PLOS ONE’s publication criteria as it currently stands. Therefore, we invite you to submit a revised version of the manuscript that addresses the points raised during the review process.

We look forward to receiving your revised manuscript.

Kind regards,

Ramona Bongelli, Ph.D.

Academic Editor

PLOS ONE

“I have read the journal's policy and the authors of this manuscript have the following competing interests: co-author Debby G. Keuken is a senior policy officer at the NVVC. Co-author Rutger J. Hassink was chair of the Quality Committee of the NVVC during the period of data-collection. Co-author José P. S. Henriques is a board member of the NVVC. The other authors have no conflicts of interest.”

Please respond by return email with your amended Competing Interests Statement and we will change the online submission form on your behalf.

Reviewers' comments:

Reviewer's Responses to Questions

**Comments to the Author**

1. Is the manuscript technically sound, and do the data support the conclusions?

Reviewer #1: Partly

Reviewer #2: Yes

2. Has the statistical analysis been performed appropriately and rigorously? 

Reviewer #1: Yes

Reviewer #2: Yes

3. Have the authors made all data underlying the findings in their manuscript fully available?

Reviewer #1: Yes

Reviewer #2: Yes

4. Is the manuscript presented in an intelligible fashion and written in standard English?

Reviewer #1: Yes

Reviewer #2: Yes

5. Review Comments to the Author

Reviewer #1: Thank you for the opportunity to review this manuscript. The topic is interesting and have potential significance for the improvement of cardiologists' professional fulfillment. However, something for authors to consideration is attached, see below.

Abstract

The first sentence of the conclusion overstates the results of this study, please revise it.

Introduction

Page 4, line 93, “ant” should be “and”.

Methods

1. Please describe how the sample size is calculated, and what the inclusion and exclusion criteria are.

2. I wonder the criteria the author included demographic characteristics, what was the reason to select sex, age, years of experience as a cardiologist (year of first registration) and type of institution?

3. Without univariate analysis, why the author only included gender and years of experience as covariates from 4 demographic characteristics?

Results

1. In Table 1 and Table 2, the bottom lines were missed.

2. Is there any difference in professional fulfillment by demographic characteristics?

Discussion

1. the short titles are not appropriate.

2. The dependent variable is professional fulfillment, why the author focus on the score for self-kindness, not the score for professional fulfillment?

3. the discussion is need to be expanded deeply.

Conclusions

1. Please move the first sentence to discussion section.

2. Again, why the second sentence focused on self-kindness, not professional fulfillment.

Reviewer #2: Thank you for submitting your research paper for PLOS. While this paper is well-written, there are a few issues that need to be fulfilled. This includes:

- The introduction can be improved. Explicitly introduce research questions / research objectives towards the end of the introduction, as bullet points, before sharing how the remaining article is structured.

- The introduction must include the importance of this work can be highlighted at the end of the introduction. Also, the novelty of this paper should be further justified by highlighting main contributions to the existing literature. The authors need to provide a strong argument here to show their novelty contributed to the field, the entire readership and the research community.

- In the methodology and research design section please define the population of the study and sampling technique adopted. Also please justify why your chosen sampling technique and sample size are appropriate.

- The authors are suggested to have a stronger discussion on the findings before the conclusion section. Discussion should ideally have 2-3 subsections on Contributions to literature, Implications for practice and Limitations and future research directions.

6. PLOS authors have the option to publish the peer review history of their article (what does this mean?). If published, this will include your full peer review and any attached files.

Reviewer #1: No

Reviewer #2: **Yes: **Ahmad Samed Al-Adwan

---

## [Author Response · Author response to Decision Letter 0]

27 Mar 2023

Dear editor/ editorial staff,

Please find our responses to specific reviewer and editor comments in the table in our rebuttal letter - the file named 'Response to Reviewers'. 

Kind regards, also on behalf of the co authors,

Rosa Bogerd

---

## [Editor Report · Decision Letter 1]

3 Apr 2023

The relationship between physicians’ self-kindness and professional fulfillment and the mediating role of personal resilience and work-home interference: A cross-sectional study

PONE-D-22-32505R1

Dear Dr. Bogerd,

We’re pleased to inform you that your manuscript has been judged scientifically suitable for publication and will be formally accepted for publication once it meets all outstanding technical requirements.

Kind regards,

Ramona Bongelli, Ph.D.

Academic Editor

PLOS ONE

Additional Editor Comments (optional):

Dear Authors

thank you for responding to the reviewers' requests in a timely manner.

I would ask you, before the final submission of the article, to do a further editing check. There are still a few small things to fix (especially near the bibliographical references). I am referring to small typographical errors.

In line 230 you claim to have used 'years of registration' because it is more clinically relevant. Can you explain why in the main text of the article? If possible, can you add bibiographical references to support your claim?
---

## [Editor Report · Acceptance letter]

13 Apr 2023

PONE-D-22-32505R1 

The relationship between physicians’ self-kindness and professional fulfillment and the mediating role of personal resilience and work-home interference: A cross-sectional study 

Dear Dr. Bogerd:

I'm pleased to inform you that your manuscript has been deemed suitable for publication in PLOS ONE. Congratulations! Your manuscript is now with our production department. 

Kind regards, 

on behalf of

Professor Ramona Bongelli 

Academic Editor

PLOS ONE